# Comparison between Intraocular Pressure Profiles over 24 and 48 h in the Management of Glaucoma

**DOI:** 10.3390/jcm12062247

**Published:** 2023-03-14

**Authors:** Philip Keye, Daniel Böhringer, Alexandra Anton, Thomas Reinhard, Jan Lübke

**Affiliations:** 1Eye Center, Medical Center–University of Freiburg, Faculty of Medicine, University of Freiburg, 79106 Freiburg, Germany; 2ADMEDICO Augenzentrum, Fährweg 10, 4600 Olten, Switzerland

**Keywords:** glaucoma, IOP, diurnal, nocturnal, fluctuation, variation

## Abstract

(1) Background: Due to significant variation, sporadic IOP measurements often fail to correctly assess the IOP situation in glaucoma patients. Thus, diurnal-nocturnal IOP profiles can be used as a diagnostic tool. The purpose of this study is to determine the additional diagnostic value of prolonged IOP profiles. (2) Methods: All diagnostic 48 h IOP profiles from a large university hospital, between 2017 and 2019, were reviewed. Elevated IOP > 21 mmHg, IOP variation > 6 mmHg and nocturnal IOP peaks were defined as IOP events of interest and counted. The analysis was repeated for the first 24 h of every IOP profile only. The Chi^2^ test was used for statistical analysis. (3) Results: 661 IOP profiles were included. Specifically, 59% of the 48 h IOP profiles revealed IOP values above 21 mmHg, and 87% showed IOP fluctuation greater than 6 mmHg. Nocturnal peaks in the supine position could be observed in 51% of the patients. In the profiles censored for the first 24 h, the fractions were 50%, 71% and 48%, (*p* < 0.01, *p* < 0.01 and *p* = 0.12) respectively. (4) Conclusions: the 48 h IOP profiles identified more patients with IOP events of interest than the 24 h IOP profiles. The additional diagnostic value must be weighed against the higher costs.

## 1. Introduction

Glaucoma is characterized by a progressive loss of retinal ganglion cells and represents one of the leading causes of blindness worldwide [1,2,3]. Several risk factors for the emergence and progression of glaucoma have been described, including older age, a family history of glaucoma, exfoliation, lower systolic blood pressure and elevated intraocular pressure (IOP) [4,5]. Among these, elevated IOP is the most important, since lowering the IOP has been shown to be the only therapeutic approach to reduce the risk of progressive loss of ganglion cells, and therefore, visual field loss [6]. In healthy humans, the IOP usually ranges from 10 to 21 mmHg and shows a significant fluctuation over a period of 24 h [7,8]. The role of IOP fluctuation has been the subject of debate in recent years, with some studies suggesting that intraocular IOP fluctuation is an independent risk factor for the progression of glaucoma, and some studies suggesting otherwise [9]. For example, Tajunisah et al. reported a mean IOP amplitude of 6 mmHg in glaucoma suspects compared to 4 mmHg in healthy eyes [10]. It has been shown that IOP frequently peaks at night. This is usually attributed to a supine sleeping position and the circadian rhythm [11].

With IOP being the only modifiable risk factor for visual field loss in glaucoma patients, it is of utmost importance that high IOP values are detected and that those patients who should be considered for IOP-lowering therapeutic measures are identified. IOP is usually measured within normal office hours in an outpatient setting. However, it has been shown that sporadic IOP measurements often fail to reproduce IOP mean values, due to IOP fluctuation and the diurnal and nocturnal changes [12]. In many places, glaucoma patients with suspected progressive visual field loss and seemingly normal IOP values are hospitalized, and the IOP is measured repeatedly in order to approach the true mean IOP and rule out considerable IOP peaks that might contribute to the progression of glaucomatous damage [13]. Having knowledge of the individual IOP situation of a patient is valuable when it comes to deciding what therapeutic measures are best suited for that particular patient. The diagnostic value is largely independent of concurrent glaucoma medication or past glaucoma surgery, because IOP remains the only truly modifiable risk factor across all stages of glaucoma. If the IOP profile reveals an uncontrolled IOP situation as the cause of glaucoma progression, lowering the IOP is necessary irrespective of present glaucoma medication or past glaucoma surgery. It has been shown that diurnal pressure patterns show poor repeatability both in healthy controls and glaucoma patients [14,15]. Fischer et al. reported IOP profile data of a cohort of 80 patients who received a diurnal and nocturnal IOP profile, suggesting that both the maximum and mean IOP differed between day 1 and day 2 of the profile [16]. In recent years, a lot of effort has been put into the development of continuous IOP monitoring devices and the advances in this field are substantial. The desire for continuous IOP monitoring, e.g., over 24 h, arose from the limitations of singular IOP measurements: IOP fluctuation and IOP peaks could be detected more reliably without creating an artificial measuring situation, ideally at the patient’s home. Both invasive and non-invasive approaches have been tested. Intraocular sensors (e.g., EYEMATE, Implandata Ophthalmic Products GmbH, Hannover, Germany) must be implanted into the eye, while most non-invasive approaches incorporate different designs for corneal contact lenses (e.g., Triggerfish CLS, Sensimed AG, Lausanne, Switzerland). Both technologies face difficulties. The implantation of an intraocular sensor carries a disproportionate risk of intraocular infections. Additionally, combined cataract surgery is usually required. Non-invasive contact lens technologies, on the other hand, are prone to measuring inaccuracies due to confounding factors, such as corneal curvature [17]. Due to these challenges, continuous IOP monitoring is not yet broadly incorporated into the clinical routine of glaucoma management, and therefore, “classical” measuring of the IOP remains of major importance.

In this study, we aim to determine the additional value of IOP profiles over 48 h, compared to IOP profiles over 24 h, regarding the ability to identify patients with elevated IOP values and significant IOP fluctuation. Additionally, we analyze whether potential nocturnal peaks are more frequently detected when measuring the IOP over 48 h. Since our goal was to obtain findings that are as universally valid as possible and of as much practical use as possible, all our data are based on a “real world” cohort of hospitalized patients from a large eye hospital, comprising different underlying ophthalmic diseases from the glaucoma family.

## 2. Materials and Methods

This retrospective mono-center cohort study was approved by the local ethics committee (vote no. 21-1184).

We manually reviewed our hospital database from the years 2017 to 2019 and identified all patients who had been admitted for an IOP profile, as part of the diagnostic routine. The reason for admission was mainly confirmed glaucoma with suspected progression of visual field loss under therapy (conservative mono- or multitherapy and/or surgical therapy) or, less frequently, suspected glaucoma with normal pressure values at outpatient presentation. Glaucoma diagnosis was established by a glaucoma specialist in accordance with the latest guidelines by the European Glaucoma Society and based on clinical examination, visual field examination and optical coherence tomography (OCT) measurements. Clinical signs, such as optic disc hemorrhage or deterioration of visual acuity due to glaucoma, were considered as signs of progression. Analysis tools from the OCT device manufacturer (Heidelberg Engineering, Heidelberg, Germany) were used to detect progression in the OCT measurements and trend-based analysis tools were incorporated to detect progression of visual field loss. We did not further distinguish between different types of glaucoma. The IOP profile usually starts on the first day of hospitalization at noon and is conducted over 24 or 48 h. The time points of IOP measurements were 12:00, 16:00, 20:00, 00:00 and 07:00. This resulted in a total of 10 IOP measurements over the period of 48 h and 5 over the period of 24 h. The 12:00, 16:00 and 20:00 measurements were performed using Goldmann applanation tonometry (GAT) in a sitting position; the 00:00 and 07:00 measurements were performed using a handheld contact tonometer (iCare pro tonometer, iCare Finland Oy) in a supine position. Measurements with the iCare device were repeated five times. The IOP values were only accepted if the device’s integrated quality control approved the measurement. At the start of every IOP profile, a comparison measurement was performed to see if the iCare IOP values matched the GAT IOP values. If differences were found, these were incorporated into the iCare measured values. For organizational reasons, not all measurements were performed by the same investigator. Patients were instructed to adhere to their usual sleeping schedule. In this study, only IOP profiles over 48 h were considered for further analysis. IOP profiles with missing IOP values were excluded from the analysis. Of all the IOP profiles, 661 profiles matched these criteria and were included in the analysis.

For this study, we operationalized clinically meaningful IOP events according to the following criteria:At least one IOP measurement over 21 mmHg;IOP fluctuation over 6 mmHg, over the course of the IOP profile;The IOP maximum in one of the nocturnal measurements in the supine position.

In the first step, we analyzed what percentage of all 48 h IOP profiles met any of these criteria. In the second step, the analysis was repeated for the same dataset, albeit censored for the first 24 h only. The data were analyzed using the R platform [18]. A Chi^2^ test was used to compare the event rates between the groups. The alpha level was set to 0.05. We did not correct for multiple testing.

## 3. Results

Approximately 43% of the IOP profiles were derived from male patients and 57% from female patients. The mean age at the time point of the IOP profile was 64 years (56/64/75 quartiles). The underlying diagnosis was primary or secondary open-angle glaucoma (including pseudoexfoliative glaucoma, normal-tension glaucoma and pigmentary glaucoma) in 49% of the patients. In 9%, the diagnosis was glaucoma suspect or ocular hypertension. The remaining 42% consisted of glaucoma secondary to eye trauma, inflammation, other eye disorders, drugs (e.g., corticosteroids) and other, not further specified, glaucomas.

The share of IOP profiles that showed an IOP above 21 mmHg at least once within 48 h was 59%. When only the first 24 h of every IOP profile were considered, this percentage was 50%. This means that 9% of the eyes showed elevated IOP values only during the second day of the IOP profile. The difference was statistically significant (*p* < 0.01).

In 87% of all the eyes, the IOP showed a fluctuation of above 6 mmHg over the period of 48 h, whereas this was the case in 71% of the profiles when only the first 24 h were considered. This result was statistically significant (*p* < 0.01).

We analyzed whether the peak IOP of every IOP profile was measured in one of the nocturnal measurements (00:00 or 07:00, supine position). This was the case in 51% of the profiles over the period of 48 h, and in 50% of the cases when only the first 24 h were considered. The difference was not statistically significant (*p* = 0.12).

All results are shown in Table 1.

## 4. Discussion

In this study, we present data from a comparatively large number of IOP profiles that were collected for diagnostic purposes from our hospital. While to date there is little evidence that IOP profiles are important to limit visual field loss in glaucoma, recent literature tends to recommend IOP measurements outside normal office hours to identify patients at risk [13,16,19]. However, the implementation of routine IOP measurements outside normal office hours can be challenging from a practical perspective and certain biases, such as different examiners or interruptions to the patient’s normal routine, can hardly be avoided. In Germany, inpatient IOP profiles are incorporated into the diagnostic process, for example, when glaucoma progression is suspected, while the IOP is repeatedly within the therapeutic range at outpatient visits. Those IOP profiles are usually conducted over the course of a 24 or 48 h inpatient stay. For the IOP profile to meet its intended purpose, it is crucial to gather enough data, but the benefit of longer measurement periods must be weighed against the additional costs. In our analysis, 9% of the eyes showed IOP values above 21 mmHg exclusively in the second 24 h of measurement. This observation supports the already published data on the poor reproducibility of diurnal IOP patterns. However, missing and potentially relevant IOP events might bear the risk of undertreatment. Regarding IOP fluctuation, our data suggest that IOP profiles over 48 h will identify more patients with significant short-term IOP variation. Taken together, our analysis adds to the published data on IOP profiles over 48 h being more reliable than IOP profiles over 24 h, regarding the detection of IOP peaks and elevated IOP means [15,16]. Our data quantify the additional benefit of a longer IOP profile regrading certain IOP events. Importantly, our study does not allow conclusions to be drawn on the optimal duration of an IOP profile, nor on the general value of this diagnostic tool. Many of the disadvantages and structural biases of traditional IOP profiles could be overcome by using technologies that allow for continuous IOP monitoring over a certain time frame, such as implantable devices or non-invasive approaches. The ideal device would need to gather as many accurate IOP measurements as possible in a certain time frame, be safe for use on patients and be cost efficient. However, due to technical issues with measuring accuracy, safety concerns (e.g., infections) and high costs, none the present technologies have been brought into broader use in the management of glaucoma.

The socio-economic aspects of diagnostic efforts should also be considered. The global prevalence of glaucoma is expected to rise in the near future, and by the year 2040 nearly 112 million people could be affected by the disease [3]. In Germany, a large population-based prospective cohort study suggested a glaucoma prevalence of 1.44% in 2018 [20]. Based on a population with an estimated 83 million people, this results in roughly 1.2 million manifest glaucoma cases. The economic burden of glaucoma is significant. The direct medical costs include medication, consultations and hospital visits. Examples of direct non-medical costs are transportation and public financial aid programs for the blind. Moreover, indirect costs such as a loss of productivity of patients and the need for caregivers add to the total costs [21]. Direct medical costs alone were reported to exceed EUR 1000 per year in western European countries at the beginning of the century [22,23]. More recent data from the US suggested annual direct costs of approximately USD 2200 for stage 5 glaucoma patients [24]. In Germany, the second night of an inpatient IOP profile accounts for approximately EUR 300, which has to be covered by the patient’s health insurance. Given the diagnostic advantage of 48 h IOP profiles that our data suggest, this expense might be well invested if follow-up costs resulting from disease progression can be avoided.

By design, this study has certain limitations. We cannot determine whether the incorporation of IOP profiles per se improves long-term glaucoma outcomes. Furthermore, IOP events were operationalized in a simple manner irrespective of the form of glaucoma or anti-glaucomatous therapy. This ignores the possibility that different forms of glaucoma may differ significantly in the nature of their IOP variation. However, the detection of an uncontrolled IOP is equally important in all types of primary and secondary glaucoma and higher IOP fluctuation has been discussed as a risk factor in primary and secondary open-angle glaucoma, as well as in normal-tension glaucoma. Due to the reimbursement criteria for health insurance, our cohort predominantly consisted of patients under extensive anti-glaucomatous therapy and patients who had already experienced surgical intervention to lower their IOP. The early stages of glaucoma might show different IOP patterns. The difference of 16% of patients experiencing IOP fluctuation between the 24 and 48 h analysis should be viewed with caution because nocturnal measurements are performed with a handheld tonometer, whereas diurnal measurements are performed using GAT. Additionally, not all GAT measurements were conducted by the same examiner.

## 5. Conclusions

On the basis of our data, gathered from a “real world” mix of glaucoma patients and glaucoma suspects, we tend to recommend a time frame of 48 h rather than 24 h when IOP profiles are used as a diagnostic tool, especially in cases where IOP values were normal during the first 24 h of the profile. However, larger and randomized clinical trials are warranted to evaluate the general diagnostic value of IOP profiles.

## Figures and Tables

**Table 1 jcm-12-02247-t001:** Fractions of IOP profiles that met the predefined criteria for the 24 h and the 48 h group.

	After 24 h	After 48 h	Chi^2^ Test
IOP_max_ > 21 mmHg	50%	59%	*p* < 0.01
IOP fluctuation > 6 mmHg	71%	87%	*p* < 0.01
IOP peak in supine position	48%	51%	*p* = 0.12

## Data Availability

The data are available on reasonable request.

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
