# Peer review of "Comparison between Intraocular Pressure Profiles over 24 and 48 h in the Management of Glaucoma"

_jcm, 2023, doi:10.3390/jcm12062247_

Round 1

Reviewer 1 Report

Authors look at very important aspect in glaucoma diagnosis and glaucoma. I still have the following comments:

a) aim of this study is to assess diagnostic value of IOP measurements over 24 or 48 hrs. I came then to realize that there were patients who were already on medications and even surgery. what the authors then mean with the diagnostic value. 

b) please update the readers about recent technologies for continuous IOP measurments and what potential benefits from such approach 

c) methods & Results need to be thoroughly presented.

- How many were newly diagnosed?  

- How many were on treatment and hence admitted for monitoring of therapy responsiveness

- Were all those patients examined by university staff having same approach glaucoma or referred from private specialists? This take me to another concern. what criteria and hence tests you used for glaucoma diagnosis particularly for the newly diagnosed?

- Tonometer measurements with iCare: which device, repetitions of measurements.

- IOP fluctuations patterns: average time of the day. Since you included already treated patients, does this pattern differ from the newly diagnosed participants. 

- Repeated measurements anova should be done to see clearly the differences between IOP levels and drawing appropriate conclusions with the following factors: 

--  IOP measuring device

-- Time: day and night

-- short or long profile: of 24 vs 48 hours measurements

-- Treatment: of IOP treatments vs no treatments

-- if possible type of glaucoma. 

In short, introduction and discussion should include recent advances in the field of IOP measurements. methods need further clarification with definition/criteria/demographic charachterestics of glaucoma groups and analysis should include all factors that influence the IOP profiles. 

Reviewer 2 Report

The manuscript entitled “Comparison between intraocular pressure profiles over 24 and 48 hours in the management of glaucoma” by Philip et al  compared intraocular pressure in glaucoma for 24 and 48 hours, finding that 48-hour IOP profiles identify more patients with IOP events of interest than 24-hour IOP profiles. I have several comments.

“With elevated IOP being the only treatable risk factor for visual field loss in glaucoma patients”. This is not true because Normal tension glaucoma (NTG) is a common form of primary open-angle glaucoma (POAG) in which there is no measured elevation of the intraocular pressure (IOP).

Your admission criteria include: 

1.    The presence of glaucoma with the suspected progression of visual field loss under intensive therapy 

2.    Suspected glaucoma with normal pressure values at outpatient presentation. 

For point 1, how do you define suspected progression? Do you have solid data to support this claim?

For point 2, Again suspected. Since you mentioned the reality of the fluctuation of IOP, how do you define normal pressure at one-time point: outpatient presentation?

Why you did not further distinguish between different types of glaucoma?

“Time points of IOP measurements are 12:00, 16:00, 20:00, 00:00 and 07:00. This results in a total number of IOP measurements of 10 over the period of 48 hours and 5 over the period of 24 hours. The 12:00, 16:00 and 20:00 measurements are performed using Gold-mann applanation tonometry (GAT) in a sitting position, the 00:00 and 07:00 measurements are performed using a handheld contact tonometer (iCare tonometer, iCare Finland Oy) in a supine position.”

Will patients mind being examined at 00:00? Will the readout and consistency differences happen between GAT and handheld contact tonometer?

What is the ratio between male and female patients?

Round 2

Reviewer 1 Report

Thanks, the Authors have addressed my comments!